# Bacterial Pulmonary Co-Infections on ICU Admission: Comparison in Patients with SARS-CoV-2 and Influenza Acute Respiratory Failure: A Multicentre Cohort Study

**DOI:** 10.3390/biomedicines10102646

**Published:** 2022-10-20

**Authors:** Grégoire Delhommeau, Niccolò Buetti, Mathilde Neuville, Shidasp Siami, Yves Cohen, Virginie Laurent, Bruno Mourvillier, Jean Reignier, Dany Goldgran-Toledano, Carole Schwebel, Stéphane Ruckly, Etienne de Montmollin, Bertrand Souweine, Jean-François Timsit, Claire Dupuis

**Affiliations:** 1Service de Pneumologie, CHU Gabriel Montpied, 63000 Clermont-Ferrand, France; 2Unité Mixte de Recherche (UMR) 1137, IAME, Université Paris Cité, 75018 Paris, France; 3Infection Control Program and WHO Collaborating Centre on Patient Safety, Faculty of Medicine, University of Geneva Hospitals, 1205 Geneva, Switzerland; 4Polyvalent Intensive Care Unit, Hôpital Foch, 92150 Suresnes, France; 5General Intensive Care Unit, Sud Essonne Hospital, 91150 Etampes, France; 6Intensive Care Unit, University Hospital Avicenne, AP-HP, 93000 Bobigny, France; 7Polyvalent Intensive Care Unit, André Mignot Hospital, 78150 Le Chesnay, France; 8Medical Intensive Care Unit, University Hospital of Reims, 51100 Reims, France; 9Medical Intensive Care Unit, University Hospital of Nantes, 44000 Nantes, France; 10Medical and Surgical Intensive Care, Montfermeil Hospital, 93370 Montfermeil, France; 11Medical Intensive Care Unit, University Hospital Grenoble-Alpes, 38000 Grenoble, France; 12Medical and Infectious Diseases Intensive Care Unit, Bichat Hospital, AP-HP, 75018 Paris, France; 13Medical Intensive Care Unit, University Hospital Gabriel Montpied, 63000 Clermont-Ferrand, France; 14Unité de Nutrition Humaine, INRAe, CRNH Auvergne, Université Clermont Auvergne, 63000 Clermont-Ferrand, France

**Keywords:** SARS-CoV-2, *influenza*, bacterial pulmonary co-infection, outcome, intensive care, ventilator-associated pneumonia

## Abstract

Background: Few data are available on the impact of bacterial pulmonary co-infection (RespCoBact) during COVID-19 (CovRespCoBact). The aim of this study was to compare the prognosis of patients admitted to an ICU for *influenza* pneumonia and for SARS-CoV-2 pneumonia with and without RespCoBact. Methods: This was a multicentre (*n* = 11) observational study using the Outcomerea© database. Since 2008, all patients admitted with *influenza* pneumonia or SARS-CoV-2 pneumonia and discharged before 30 June 2021 were included. Risk factors for day-60 death and for ventilator-associated-pneumonia (VAP) in patients with *influenza* pneumonia or SARS-CoV-2 pneumonia with or without RespCoBact were determined. Results: Of the 1349 patients included, 157 were admitted for *influenza* and 1192 for SARS-CoV-2. Compared with the *influenza* patients, those with SARS-CoV-2 had lower severity scores, were more often under high-flow nasal cannula, were less often under invasive mechanical ventilation, and had less RespCoBact (8.2% for SARS-CoV-2 versus 24.8% for *influenza*). Day-60 death was significantly higher in patients with SARS-CoV-2 pneumonia with no increased risk of mortality with RespCoBact. Patients with *influenza pneumonia* and those with SARS-CoV-2 pneumonia had no increased risk of VAP with RespCoBact. Conclusions: SARS-CoV-2 pneumonia was associated with an increased risk of mortality compared with *Influenza* pneumonia. Bacterial pulmonary co-infections on admission were not associated with patient survival rates nor with an increased risk of VAP.

## 1. Introduction

Due to the scale of its spread, severe acute respiratory syndrome coronavirus 2 (SARS-CoV-2) was rapidly compared with the *influenza* virus, in particular with regard to pulmonary bacterial co-infection on admission (RespCoBact).

Respiratory co-infection on admission in patients with *influenza* pneumonia is known to be frequent—from 19% to 47% depending on the study [1,2,3]. Such co-infections are also reported to be associated with an added risk of mortality, which varies according to the type of documented pathogen and the presence of co-morbidities [2] (Appendix A).

For patients with SARS-CoV-2 pneumonia admitted to an intensive care unit (ICU), data on these co-infections are scarce. A review of 10 studies published in 2021 [4] found that 4% of patients hospitalised for SARS-CoV-2 pneumonia had a documented co-infection on admission. The main pathogens were *Staphyloccocus aureus*, *Streptococcus pneumoniae*, and *Haemophilus influenzae*. The main risk factors for co-infection were age, chronic renal failure, heart failure, and diabetes. Co-infection was associated with an increased length of hospital stay and increased in-hospital mortality, but these findings were not reported in all studies and were heterogeneous between studies.

Few studies directly compared the impact of these co-infections in patients with *influenza* pneumonia and in patients with SARS-CoV-2 pneumonia. The most important study was published in 2021 [5] and included 1050 patients. The prevalence of bacterial co-infections was significantly lower in patients with SARS-CoV-2 pneumonia than in those with *influenza* pneumonia (9.7% vs. 33.6%). Bacterial co-infection tended to be associated with an increased risk of mortality at D28 in SARS-CoV-2 pneumonia but not in *influenza* pneumonia.

The results from one study to another thus seem heterogeneous and for several studies were based on a small cohort. Furthermore, risk factors of co-infections are scarcely reported. Due to the heterogeneity of the results and the paucity of the data, those results should be confirmed.

In that context, the purpose of this study was to analyze the epidemiology and prognostic impact of RespCoBact in ICUs in patients with SARS-CoV-2 pneumonia and *influenza* pneumonia.

## 2. Materials and Methods

This was a French multicentre (*n* = 11) prospective observational study using data from the Outcomerea© database.

### 2.1. Database

In compliance with French law, the Outcomerea© database was approved by the Comité Consultatif sur le Traitement de l’Information en matière de Recherche dans le domaine de la Santé (CCTIRS) and by the Commission Nationale Information et Libertés (CNIL, No. 8999262). The database protocol was submitted to the research ethics committee (IRB) of the University of Clermont-Ferrand, which agreed that there was no requirement for informed consent. Information was given to the patient or their family.

### 2.2. Study Population

All patients over 18 years of age admitted to one of the ICUs in the Outcomerea© group could be included in the Outcomerea© database. For our study, we included patients with SARS-CoV-2 or *influenza* pneumonia admitted to the ICU after 1 January 2008 and ending their ICU stay before 30 June 2021. They had to have severe SARS-CoV-2 or *influenza* pneumonia with a positive SARS-CoV-2 or *influenza* RT-PCR.

Patients without a complete follow-up and patients infected with both *influenza* and SARS-CoV-2 were excluded.

### 2.3. Data Collection

Data that were collected prospectively from admission to the ICU included demographic data; chronic diseases including respiratory, cardiac, renal, and hepatic comorbidities according to the Knaus classification, SAPS II (Simplified Acute Physiology Score), and SOFA (Sequential Organ Failure Assessment) severity indices; and treatments received on admission including different antibiotic therapies and corticosteroid and immunomodulatory treatments. Other variables included with the clinical and biological variables were ventilation modalities; other organ support; lung diseases on admission and nosocomial infections arising in the ICU, including ventilator-associated pneumonia (VAP); and length of stay and patient outcome at 60 days.

### 2.4. Definitions

RespCoBact in the ICU was defined as the presence of a community-acquired or hospital-acquired bacterial pneumonia associated with *influenza* pneumonia or SARS-CoV-2 pneumonia during the ICU admission.

The presence of RespCoBact was defined as the presence of radiological and/or scanographic condensation, bacteriological documentation (a positive quantitative culture of lower respiratory tract samples collected as recommended (bronchoalveolar lavage, >10^4^ CFU/mL, plugged telescoping catheter, >10^3^ CFU/mL, endotracheal aspirate, >10^6^ CFU/mL)), and/or presence of positive antigenuria, as defined by the European Centre for Disease Control and Prevention [6,7]. RespCoBact was deemed community-acquired if diagnosed within the first 48 h of hospital admission and hospital-acquired if diagnosed after 48 h. If bacterial pneumonia occurred at least 2 days after intubation, it was classified as VAP [8]. The risk period for VAP begins at 48 h after intubation and lasts until removal of the tracheal tube and weaning from the invasive ventilation, so it ends with extubation.

Causal agents of RespCoBact were defined as multidrug-resistant (MDR) on the following criteria: methicillin-resistant *Staphylococcus aureus* (MRSA), extended-spectrum beta-lactamase (ESBL)-producing *Enterobacteriaceae*, AmpC-producing *Enterobacteriaceae*, and *Pseudomonas aeruginosa* resistant to ticarcillin and/or imipenem and/or ceftazidime.

Invasive mechanical ventilation was defined as continuous mechanical ventilation using either an endotracheal tube or a tracheostomy tube. The other oxygenation modalities included face mask ventilation modes with noninvasive positive-pressure ventilation (NIPPV), continuous positive airway pressure (CPAP), and high-flow nasal cannula (HFNC). These techniques could be continuous or deferred.

### 2.5. Statistical Analysis

Patient characteristics were expressed as number and percentage for categorical variables and median and interquartile range for continuous variables. Comparisons were made using Fisher’s test for categorical variables and Wilcoxon’s test for continuous variables.

Univariate and multivariate analyses with a Cox survival model were performed to investigate the independent risk factors for death at 60 days. Variables reaching a *p*-value < 0.2 in the univariate analysis were tested in the multivariate analysis. Variables were then entered using a backward procedure and only those with a *p*-value < 0.05 were retained. In a similar way, factors associated with the occurrence of VAP among the patients at risk of VAP were sought. The Fine–Gray subdistribution hazard model was used while taking into account death and discharge alive from the ICU as competitive risks.

Finally, risk factors for respiratory co-infections were sought using a univariable logistic regression analysis.

## 3. Results

### 3.1. Comparison of Patients with Influenza Pneumonia versus SARS-CoV-2 Pneumonia

During the study period, 1463 patients were admitted to one of the Outcomerea© group ICUs with a diagnosis of SARS-CoV-2 or *influenza*-related pneumonia. Data from 1349 of these patients, including 157 patients with *influenza* and 1192 patients with SARS-CoV-2, were analysed (Figure 1). The main characteristics are reported in Table 1. Most of the patients in the *influenza* group were admitted before the COVID pandemic and after 2012 (*n* = 135 (84.7%)); 41.8% of the patients with SARS-CoV-2 were admitted to the ICU during the first wave of the pandemic (February–May 2020).

These two populations (SARS-CoV-2 pneumonia versus *influenza* pneumonia) differed in several ways. The patients in the SARS-CoV-2 pneumonia group were older and more often male. More were obese and more had cardiovascular comorbidities. In contrast, patients in the *influenza* pneumonia group had more comorbidities, especially chronic respiratory failure and immunosuppression.

The time from hospital admission to ICU admission was shorter for patients with *influenza* pneumonia than for patients with SARS-CoV-2 pneumonia (*p* < 0.01). On admission, the SAPS II and SOFA severity scores were significantly higher in the *influenza* pneumonia group, with more use of mechanical ventilation. Patients in the SARS-CoV-2 pneumonia group had more severe hypoxaemia and were predominantly managed using HFNC.

Regarding the treatments received on admission, patients with SARS-CoV-2 pneumonia more often received corticosteroids. Approximately 60% of the patients were receiving antibiotic therapy on admission to the ICU, with most using third-generation cephalosporins and macrolides in the SARS-CoV-2 pneumonia group and betalactam antibiotics with inhibitors, fluoroquinolones, or antistaphylococcal treatment in the *influenza* pneumonia group.

RespCoBact was more frequently observed in the *influenza* pneumonia group (24.8%) than in the SARS-CoV-2 pneumonia group (8.2%); *p* < 0.01. The proportion of community-acquired infections was higher in the patients with *influenza* pneumonia (84.6%) than in those with SARS-CoV-2 pneumonia (58.2%); *p* < 0.01.

During their ICU stay, the patients with *influenza* were more often intubated (*p* < 0.01). However, the duration of mechanical ventilation did not differ from that for patients with SARS-CoV-2 pneumonia. The patients with SARS-CoV-2 pneumonia were more often placed in a prone position (*p* = 0.07). There was no significant difference in the use of ECMO between the two groups (*p* = 0.19). Patients with *influenza* and those with SARS-CoV-2 pneumonia had nearly equivalent VAP rates (14.6% versus 17.5%, *p* = 0.37).

### 3.2. Comparison between Influenza Pneumonia and SARS-CoV-2 Pneumonia with and without Respiratory Bacterial Co-Infection at ICU Admission (Table 2)

#### 3.2.1. Analysis of the Subgroup of Patients with Influenza Pneumonia

A subgroup analysis (Table 2) showed that subjects co-infected on admission were more severely ill and had higher SOFA scores. The use of vasopressors, renal replacement therapy, oseltamivir, and antibiotic therapy was more frequent in patients with co-infections. During their stays in the ICU, the use of ECMO and vasopressors was greater. There was no difference in the length of stay or mortality.

**Table 2 biomedicines-10-02646-t002:** Comparison of patients with influenza or SARS-CoV-2 pneumonia with and without respiratory bacterial co-infection on admission.

Variables (*n* (%)/Median [IQR])	1. No FluRespCoBact	2. FluRespCoBact	*P* 1|2	3. No CovRespCoBact	4. CovRespCoBact	*p* 3|4	*p* All	*p* 1|3	*p* 2|4
**Number of patients**	118	39		1094	98				
**Time from hospital admission to ICU (days)**	1 [1; 3]	1 [1; 2]	0.14	2 [1; 4]	2 [1; 6]	0.33	<0.01	<0.01	<0.01
**Age (years)**	59.8 [52.1; 71.4]	61.1 [44.6; 72]	0.57	64.4 [54.3; 73.0]	64.1 [55.2; 70.2]	0.48	0.09	0.07	0.21
**Sex (male)**	71 (60.2)	22 (56.4)	0.68	790 (72.2)	74 (75.5)	0.48	<0.01	<0.01	0.03
**Body mass index, kg/m²**	27.3 [24.0; 31.7]	25.6 [21.2; 27.6]	<0.01	28.41 [25.1; 32.1]	27.54 [23.9; 34.4]	0.59	<0.01	0.11	<0.01
**Comorbidities**									
**Charlson score**	2 [1; 4]	1 [0; 2]	<0.01	1 [0; 3]	2 [0; 4]	0.02	<0.01	<0.01	0.05
**Chronic cardiovascular disease**	20 (17.0)	2 (5.1)	0.07	275 (25.1)	31 (31.6)	0.16	<0.01	0.05	<0.01
**Chronic lung disease**	37 (31.4)	12 (30.8)	0.95	123 (11.2)	10 (10.2)	0.75	<0.01	<0.01	<0.01
**Chronic kidney disease**	16 (13.6)	2 (5.1)	0.15	98 (8.96)	10 (10.2)	0.68	0.31	0.10	0.34
**Immunodepression ***	51 (43.2)	8 (20.5)	0.01	132 (12.1)	14 (14.3)	0.52	<0.01	<0.01	0.37
**Diabetes**	21 (17.8)	5 (12.8)	0.47	165 (15.1)	19 (19.4)	0.26	0.58	0.44	0.36
**Characteristics on admission**									
**SAPS II score**	39.5 [27; 51]	42 [32; 64]	0.10	33 [24; 43]	33.5 [26; 48]	0.16	<0.01	<0.01	0.02
**SOFA score**	5 [3; 7]	7 [5; 10]	<0.01	5 [4; 7]	6 [4; 8]	<0.01	<0.01	0.73	0.08
**PaO_2_/FiO_2_ (missing data = 60)**	150 [103; 238]	133 [82; 199]	0.10	107 [74; 180]	110 [75; 154]	0.66	<0.01	<0.01	0.20
**Organ support at admisison**									
**Invasive mechanical ventilation**	52 (44.1)	27 (69.2)	<0.01	304 (27.8)	41 (41.8)	<0.01	<0.01	<0.01	<0.01
**ECMO**	0	2 (5.1)	0.01	17 (1.6)	4 (4.1)	0.07	0.04	0.17	0.79
**Vasopressors**	10 (8.5)	8 (20.5)	0.04	204 (18.7)	27 (27.6)	0.03	<0.01	<0.01	0.39
**Renal replacement therapy**	4 (3.4)	5 (12.8)	0.03	35 (3.2)	10 (10.2)	<0.01	<0.01	0.91	0.66
**Corticoids**	30 (25.4)	9 (23.1)	0.77	634 (57.9)	69 (70.4)	0.02	<0.01	<0.01	<0.01
**Anti-Il-6 or anti-Il-1**	0	0	.	83 (7.6)	3 (3.1)	0.02	<0.01	<0.01	.
**Ozeltamivir**	36 (30.5)	15 (38.5)	0.36	29 (2.7)	0	0.10	<0.01	<0.01	<0.01
**Other anti-infectious treatments on admission**									
**Antibiotics**	55 (46.6)	26 (66.7)	0.03	631 (57.7)	73 (74.5)	<0.01	<0.01	0.02	0.36
**Amoxicillin/clavulanic acid**	16 (13.5)	11 (28.2)	0.13	66 (6.1)	13 (13.2)	<0.01	<0.01	<0.01	0.16
**Ureido-carboxypenicillins**	21 (17.8)	6 (15.4)	0.73	55 (5.0)	8 (8.16)	0.18	<0.01	<0.01	0.21
**3rd-generation cephalosporin**	28 (23.7)	17 (43.6)	0.02	453 (41.5)	50 (51.0)	0.07	<0.01	<0.01	0.43
**4th-generation cephalosporin**	3 (2.5)	1 (2.6)	0.99	46 (4.2)	8 (8.2)	0.07	0.19	0.38	0.23
**Macrolides**	27 (22.9)	16 (41.0)	0.03	312 (28.6)	25 (25.5)	0.52	0.16	0.19	0.07
**Aminosides**	10 (8.5)	3 (7.7)	0.88	40 (3.7)	14 (14.3)	<0.01	<0.01	0.01	0.29
**Fluoroquinolones**	8 (6.8)	5 (12.8)	0.24	43 (3.9)	11 (11.2)	<0.01	<0.01	0.14	0.79
**Anti-MSSA and anti-MRSA ^§^**	5 (4.2)	5 (12.8)	0.16	22 (2.0)	9 (9.2)	<0.01	<0.01	0.05	0.55
**Bacteraemia on admission**	2 (1.7)	4 (10.3)	0.02	31 (2.8)	11 (11.2)	<0.01	<0.01	0.47	0.87
**Organ support during stay in ICU**									
**Invasive mechanical ventilation**	61 (51.7)	28 (71.8)	0.03	484 (44.2)	62 (63.3)	<0.01	<0.01	0.12	0.34
**ECMO**	1 (0.9)	4 (10.3)	<0.01	50 (4.6)	10 (10.2)	0.01	<0.01	0.06	0.99
**Vasopressors**	12 (10.2)	9 (23.1)	0.04	347 (31.7)	43 (43.9)	0.01	<0.01	<0.01	0.02
**Renal replacement therapy**	18 (15.3)	10 (25.6)	0.14	161 (14.7)	24 (24.5)	0.01	0.02	0.88	0.89
**VAP**	14 (11.9)	9 (23.1)	0.09	187 (17.1)	22 (22.5)	0.18	0.16	0.15	0.94
**Outcome**									
**Duration of invasive mechanical ventilation (days)**	11 [4; 18]	16 [7; 23.5]	0.68	12 [6; 21]	12 [5; 18]	0.68	0.45	0.77	0.77
**Duration of oxygenation (days)**	8 [3; 17]	12 [6; 25]	0.09	8 [4; 15]	8 [4; 16.5]	0.09	0.13	0.35	0.35
**Duration of ECMO (days)**	1 [1; 1]	3.5 [2; 11.5]	0.65	10.5 [3; 16]	15 [6; 24]	0.65	0.17	0.08	0.08
**Duration of RRT (days)**	9 [5; 14]	3 [1; 14]	0.20	8 [3; 17]	5.5 [2; 14.5]	0.20	0.61	0.74	0.74
**Duration of ICU stay (days)**	6 [3; 14]	13 [7; 28]	<0.01	8 [4; 16]	9.5 [5; 18]	0.07	<0.01	<0.01	0.09
**Duration of hospital stay (days)**	15 [8; 35]	27.5 [16; 50.5]	<0.01	15 [10; 27]	15.5 [9; 30]	0.95	<0.01	0.69	<0.01
**Mortality at D60**	23 (19.5)	5 (12.8)	0.35	321 (29.3)	35 (35.7)	0.19	<0.01	0.02	<0.01

Flu: *influenza*, Cov: SARS-CoV-2, RespCoBact: bacterial respiratory co-infection on admission. * Organ transplants, AIDS, non-AIDS HIV, corticoids > 1 month or >2 mg/kg/j, chemotherapy, aplasia, or other immunodepression. ^§^ linezolid, daptomycin, vancomycin, cefazolin, or penicillin. SOFA: sequential organ failure assessment; SAPS: simplified acute physiology score; ECMO: extracorporeal membrane oxygenation; RRT: renal replacement therapy; MSSA: methicillin-susceptible *Staphylococcus aureus*; MRSA: methicillin-resistant *Staphylococcus aureus*; VAP: ventilator-associated pneumonia.

#### 3.2.2. Analysis of the Subgroup of Patients with SARS-CoV-2 Pneumonia

Patients co-infected on admission were more likely to have comorbidity, higher Charlson scores, and a higher SOFA. The use of invasive mechanical ventilation, ECMO, vasopressors, and antibiotics was more frequent in patients with CovRespCoBact (Table 2). There was no difference in the length of stay or mortality in patients with SARS-CoV-2 pneumonia with and without CovRespCoBact.

#### 3.2.3. Analysis of Patients in SARS-CoV-2 and Influenza Groups without RespCoBact

The differences between the two groups were those of the cohort (Table 1). Patients with *influenza* had longer hospital stays and lower mortality rates (Table 2).

#### 3.2.4. Analysis of Patients in SARS-CoV-2 and Influenza Groups with RespCoBact

Patients with *influenza* had a lower body mass index and higher SAPS II severity scores and were more often intubated on admission, but this difference disappeared during their stay. Patients with *influenza* had a greater length of stay than patients with SARS-CoV-2 with no difference in mortality.

### 3.3. Microbiological Description of Lung Co-Infections on Admission

Of the 1349 patients included, 137 had a co-infection and 135 co-infections were documented. The microbiological description of RespCoBact is given in Table 3.

In the patients with *influenza*, most of the pathogens identified in RespCoBact were Gram-positive cocci (GPCs), with a predominance of *S. pneumoniae* over *S. aureus*. Among the Gram-negative bacilli (GNBs), *Haemophilus* spp. was the most frequent causal agent.

In patients with SARS-CoV-2, the most common organisms identified in RespCoBact were GNBs. These were mainly *Enterobacter* spp., *E. coli*, and *Klebsiella* spp., followed by *Haemophilus* spp. Among the GPCs, *S. aureus* was proportionally found more often found *S. pneumoniae*.

Very few cases of drug resistance were reported in the cohort, with *ESBL-secreting Enterobacteriaceae* being the most common in patients with SARS-CoV-2 pneumonia only.

The pathogens identified in RespCoBact with regard to community-acquired infection differed in patients with SARS-CoV-2 pneumonia, this time showing more than 40% GPCs with *Staphylococcus* being predominant.

### 3.4. Risk Factors for Death at D60 (Figure 2; Appendix A)

After adjustment, having a bacterial co-infection on admission was not associated with an increased risk of death either in the whole cohort (aHR = 0.83 [CI95% 0.59; 1.15]; *p* = 0.26) or in the subgroups of patients with *influenza* pneumonia (aHR = 0.67 [CI95% 0.25; 1.81]; *p* = 0.43) or with SARS-CoV-2 pneumonia (aHR = 0.97 [CI95% 0.68; 1.39]; *p* = 0.88).

**Figure 2 biomedicines-10-02646-f002:**
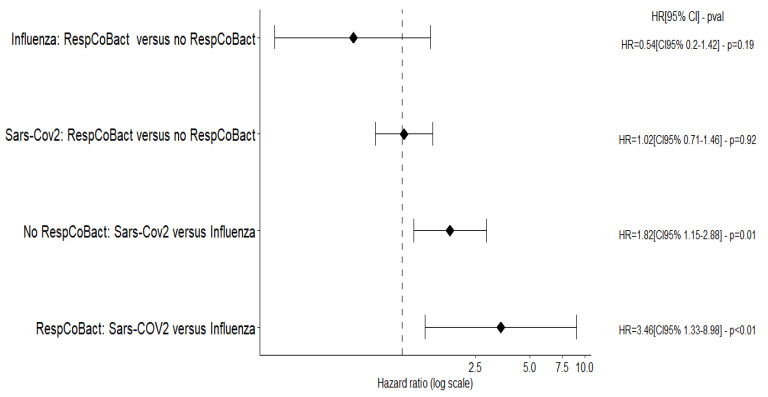
Association between SARS-CoV-2 or *influenza* and presence or absence of respiratory bacterial co-infection on admission and mortality at D60 (Cox multivariate survival model.) HR: hazard ratio at D60; RespCoBact: respiratory bacterial co-infection on admission in the Cox multivariate survival model adjusted for age, chronic cardiovascular diseases, immunodepression, PaO_2_/FiO_2_ < 150, vasopressors, extrarenal purification, and beta-lactamase inhibitors.

There was no difference in impact between GPC and GNB pneumonia. Similar results were obtained in the subgroups of immunocompromised patients and patients receiving corticosteroids on admission.

Finally, whether there was co-infection or not, patients with SARS-CoV-2 pneumonia still had a higher risk of mortality than patients with *influenza* pneumonia (Figure 2).

### 3.5. Risk Factors for VAP (Appendix A)

Co-infections were not associated with an increased risk of VAP in either SARS-CoV-2 or *Influenza* pneumonia. Viral lung diseases were not associated with the occurrence of VAP.

### 3.6. Risk Factors for Co-Infections (Appendix A)

Patients with SARS-CoV-2 pneumonia with cirrhosis (OR = 3.50 [CI95% 1.37; 8.94], *p* < 0.01) had more co-infections on admission. Immunocompromised or obese *influenza* patients had fewer co-infections on admission to the ICU (OR = 0.34 [CI95% 0.14; 0.8], *p* = 0.01; OR = 0.29 [0.1; 0.70], *p* = 0.02).

## 4. Discussion

This study found a lower prevalence of pulmonary bacterial co-infections on admission to an ICU in patients with SARS-CoV-2 pneumonia (8.2%) than in patients with *influenza* pneumonia (24.8%). In the literature, the rates of early bacterial co-infections at diagnosis of SARS-CoV-2 pneumonia at ICU admission ranged from 3% [8] to 20% [9], with a higher prevalence reported in cohorts that included a vast majority of patients on mechanical ventilation (Appendix A) [4,8,9,10,11,12,13,14,15,16]. In 254 patients admitted to seven ICUs in England during the first wave, the rate of documented bacterial coinfections was 5.5% [10]. In this study’s population, 59.5% of patients received mechanical ventilation within 24 h of admission. In a French monocentric retrospective study performed in 92 ICU patients admitted for severe COVID-19 with 83 (90%) on mechanical ventilation on admission, the rate of co-infections was 19.2 [9]. The rate of 8.2% reported in our work was very similar to the rates reported by Rouzé et al. (9.7%) [5] and Pandey et al. (8.7%) [17]. These authors also notes a higher frequency of pulmonary co-infections in patients admitted for *influenza* pneumonia: 33.6% [5] and 25% [17]. In the study by Sarton et al. [18], the difference between the two populations was less important (16% for SARS-CoV-2 pneumonia and 33% for *influenza* pneumonia) but all patients included in this study were on mechanical ventilation for at least 48 h.

The higher prevalence of pulmonary co-infections during *influenza* pneumonia may be explained in several ways. Firstly, early intubation in *influenza* pneumonia: in our study, the proportion of patients intubated on admission was 50.3% for *influenza* pneumonia and 29% for SARS-CoV-2 pneumonia. Tracheal intubation facilitated the collection of distal respiratory secretions and bacteriological documentation and so made the diagnosis of a pulmonary bacterial co-infection easier. Secondly, in *influenza*, severe bacterial co-infection may be the prime reason for early intubation rather than the *influenza* infection itself. The mechanism of hypoxia leading to admission to the ICU differs between *influenza* and SARS-CoV-2 pneumonia. In SARS-CoV-2 pneumonia, chest imaging more frequently reveals extensive ground-glass lesions associated with initial diffuse involvement of the interstitium [19,20] and microthrombi in the microcirculation [21,22]. These mechanisms probably explain a slower development of lung injury in SARS-CoV-2 pneumonia, making major hypoxaemia more progressive and better tolerated. In contrast, chest imaging during severe *influenza* reveals *influenza*-specific lesions that are often less severe and less extensive and do not by themselves justify intensive care [20]. Additional alveolar condensations secondary to bacterial infection, the onset of which is more rapid and is sometimes associated with haemodynamic instability, could explain respiratory failure and earlier intubation. It is also important to highlight progress in noninvasive oxygen therapy techniques due to the considerable rise in recent years of HFNC in the management of acute hypoxaemic respiratory failure [23]. In our study, this therapeutic evolution hindered meaningful comparison of the outcomes observed between the cohort of *influenza* patients, who were mainly admitted in 2012–2020, and the cohort of patients with SARS-CoV-2 pneumonia admitted in 2020–2021. Finally, early antibiotic therapy could also be a confounding factor, but the proportion of patients receiving antibiotic therapy on admission was not different between patients with *influenza* and those with SARS-CoV-2 pneumonia.

Regarding the bacteriological documentation of bacterial co-infections, we identified a predominance of GPCs in *influenza* pneumonia, most often *S. pneumoniae* followed by *S. aureus*. In comparison, patients with SARS-CoV-2 pneumonia had more GNB co-infections; the causal agents isolated were, in descending order of frequency, *Enterobacteriaceae* and *Haemophilus* spp. Such results were in agreement with previous findings that in *influenza* pneumonia, the two bacteria most frequently identified were first *S. Pneumoniae* and then *S. aureus* with average rates of 40% and 20%, respectively [1,2,24]. In SARS-CoV-2 pneumonia, *S aureus* was more frequently prevalent than *S. pneumoniae*, accounting for 30% vs. 20% of documented co-infections [4,9,10,16]; other major etiologic agents of co-infections were *Haemophilus* in more than 10% of the cases [4,9,13,15] and *Enterobacter* spp. in more than 25% of the cases [9,14]. In the study by Rouzé et al. [5], GPCs were identified in 58% and 72% of co-infections and GNBs in 41.8% and 27.8% in SARS-CoV-2 pneumonia and *Influenza* pneumonia, respectively. In the study by Pandey et al. [17], most co-infections were due to *S. aureus* both in SARS-CoV-2 and in *influenza* pneumonia.

The microbiological differences observed between studies and between SARS-CoV-2 and *influenza* pneumonia could reflect the different study designs (for example, Rouzé et al. [5] included only mechanically ventilated patients) or the criteria for defining co-infections (whether only those diagnosed on admission to the ICU, or including the period of prior hospitalisation, or within the first 48 h in the ICU). Another explanation could be that patients with SARS-CoV-2 pneumonia spend more time hospitalized before ICU admission than patients with *influenza* pneumonia.

We found no excess mortality associated with the presence of a bacterial respiratory co-infection in patients with *influenza* or SARS-CoV-2 pneumonia. In SARS-CoV-2 pneumonia, while some studies did not report an increased risk of mortality due to co-infections [8], two systematic reviews found a longer hospital length of stay [4] and an increased risk of death [4,11]. However, the results of these reviews should be cautiously interpreted due to the great heterogeneity of the studies. In contrast, an increased risk of mortality was observed for respiratory bacterial co-infection in patients admitted to an ICU with severe *influenza* pneumonia [2]. For instance, Rice et al. reported that co-infections due to *S. aureus* were associated with an increased risk of death [24].

In our study, we noted that compared with *influenza* patients, the increased risk of mortality observed in patients with SARS-CoV-2 with or without bacterial respiratory co-infection persisted after adjustment. This result could be partly explained by the systemic inflammation and microthrombi involved in the pathophysiology of COVID-19 [25,26]. It is also likely that in patients with *influenza*, extensive prescription of oseltamivir in the initial phase of infection attenuated the intensity of the lesions by accelerating viral clearance. Finally, some of the patients included in the SARS-CoV-2 group were managed at the beginning of the pandemic and were not receiving corticosteroids or reinforced preventive and/or curative anticoagulation, the only treatments yet shown to be of benefit in the management of SARS-CoV-2 patients [27]. However, a causal link was recently shown between the use of steroids and ICU-acquired infections [28].

We also found that the patients most at risk of co-infection on admission were those that were most severely ill and that in patients with *influenza*, comorbidities were also associated with a higher risk of bacterial pulmonary co-infection on admission.

Contrary to other studies [29], we found no increased rate of VAP in patients with SARS-CoV-2 pneumonia compared with those with *influenza* pneumonia even if among the patients at risk of VAP, the VAP prevalence was higher among patients with SARS-CoV-2 pneumonia (40%) versus those with *influenza* pneumonia (27.4%). These findings could be explained by the unbalanced between patients with SARS-CoV-2 and *influenza* pneumonia in our cohort. The occurrence of RespCoBact was also not associated with an increased risk of VAP. Contrary to our results, bacterial respiratory co-infections were already reported to be associated with an increased risk of VAP [30,31]. Our results could be explained by the severity of the underlying viral diseases and the associated prolonged duration of invasive ventilation and increased risk of death, which minimized the impact of RespCoBact upon occurrence of VAP.

The main strengths of this study were the large size of its patient population and its multicentre nature, as well as its prospective data collection using a good-quality database, which allowed an accurate study of patients with *influenza* and with COVID-19.

One limitation was the imbalance in the number of patients between those with *influenza* and those with COVID-19. In addition, the difference in recruitment periods between patients with *influenza* and those with COVID-19 and progress in noninvasive oxygen therapy strategies over time prevented any direct comparison of the impact of co-infections between the two groups.

Finally, most of the patients with COVID-19 were recruited during the first COVID-19 pandemic wave in France, in particular before the introduction of corticosteroids, which was to greatly change the evolution of COVID-19 in hypoxaemic patients

## 5. Conclusions

This multicentre observational study confirmed that bacterial pulmonary co-infections on admission to an ICU were less frequent in COVID-19 patients than in *influenza patients*.

The increased risk of mortality observed in patients with COVID-19 compared with *influenza* patients was not specifically related to the presence of a bacterial co-infection but rather to the specific mechanisms of the SARS-CoV-2 respiratory infection.

It would be of interest to repeat this study in the current period of antiviral therapies, delayed intubation strategies [32], and widespread COVID-19 vaccination.

## Figures and Tables

**Figure 1 biomedicines-10-02646-f001:**
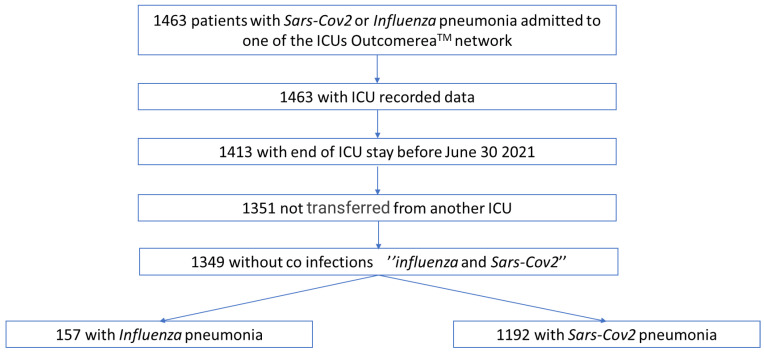
Flow diagram.

**Table 1 biomedicines-10-02646-t001:** Comparison between patients with influenza pneumonia and SARS-CoV-2 pneumonia.

Variables (*n* (%)/Median [IQR])	*Influenza* (*n* = 157)	SARS-CoV-2 (*n* = 1192)	*p*
**Time from hospital admission to ICU (days)**	1 [1; 2]	2 [1; 4]	<0.01
**Age (years)**	60.1 [51.5; 71.6]	64.4 [54.5; 72.7]	0.02
**Sex (% male)**	93 (59.2)	864 (72.5)	<0.01
**Body mass index (BMI) (kg/m²)**	26.8 [23.5; 30.9]	28.4 [25; 32.2]	<0.01
**Comorbidities**			
**Charlson score**	2 [1; 3]	1 [0; 3]	<0.01
**Chronic cardiovascular disease**	22 (14)	306 (25.7)	<0.01
**Chronic lung disease**	49 (31.2)	133 (11.2)	<0.01
**Chronic kidney disease**	18 (11.5)	108 (9.1)	0.33
**Chronic liver disease**	7 (4.5)	26 (2.2)	0.08
**Immunodepression ***	59 (37.6)	146 (12.2)	<0.01
**Diabetes**	26 (16.6)	184 (15.4)	0.71
**Characteristics on admission**			
**SAPS II score**	40 [28; 54]	33 [24; 43]	<0.01
**SOFA score**	5 [4; 8]	5 [4; 7]	0.06
**Biological data**			
**Leucocytes elts/mm^3^ (missing data = 24)**	9385 [4520; 14,700]	9000 [6660; 12,200]	0.48
**PaO_2_/FiO_2_ (missing data = 61)**	148 [95; 215]	108 [74; 177]	<0.01
**Pulmonary embolism**	0 (0)	36 (3)	0.03
**Ventilatory support on admission**			
**Invasive mechanical ventilation**	79 (50.3)	345 (29)	<0.01
**High-flow nasal cannula**	13 (8.3)	530 (44.5)	<0.01
**Continuous positive airway pressure**	30 (19.1)	138 (11.6)	<0.01
**ECMO**	2 (1.3)	21 (1.8)	0.66
**Vasopressors**	18 (11.5)	231 (19.4)	0.02
**Renal replacement therapy**	9 (5.7)	45 (3.8)	0.24
**Corticoids**	39 (24.8)	703 (59)	<0.01
**Il1 or Il6 receptor antagonists**	0 (0)	86 (7.2)	<0.01
**Lopinavir, ritonavir**	0 (0)	171 (14.4)	<0.01
**Hydroxychloroquine**	0 (0)	64 (5.4)	<0.01
**Remdesivir**	0 (0)	169 (14.2)	<0.01
**Ozeltamivir**	51 (32.5)	29 (2.4)	<0.01
**Antibiotics**	81 (51.6)	704 (59.1)	0.07
**Amoxicillin/clavulanic acid**	27 (17.2)	79 (1.7)	<0.01
**Ureido-carboxypenicillins**	27 (17.2)	63 (5.3)	<0.01
**3rd-generation cephalosporin**	45 (28.7)	503 (42.2)	<0.01
**4th-generation cephalosporin**	4 (2.5)	54 (4.5)	0.25
**Macrolides**	43 (27.4)	337 (28.3)	0.81
**Aminoglycosides**	13 (8.3)	54 (4.5)	0.04
**Fluoroquinolones**	13 (8.3)	54 (4.5)	0.04
**Anti-MSSA and anti-MRSA ^§^**	12 (7.6)	31 (2.6)	<0.01
**Co-infections on admission**			
**Bacterial pneumonia**	39 (24.8)	98 (8.2)	<0.01
**Hospital-acquired pneumonia**	6 (3.8)	41 (3.4)	0.81
**Organ support during hospital stay**			
**Invasive mechanical ventilation**	89 (56.7)	546 (45.8)	0.01
**Prone position**	20 (12.7)	293 (24.6)	<0.01
**ECMO**	5 (3.2)	60 (5)	0.31
**Vasopressors**	21 (13.4)	390 (32.7)	<0.01
**Renal replacement therapy**	28 (17.8)	185 (15.5)	0.45
**VAP**	23 (14.6)	209 (17.5)	0.37
**VAP among the patients at risk of VAP**	23/84 (27.3)	209/522 (40.0)	0.03
**Outcome**			
**Duration of invasive mechanical ventilation (days)**	12 [5; 20]	12 [6; 21]	0.77
**Duration of ECMO (days)**	3 [1; 4]	11 [4; 16.5]	0.08
**Duration of RRT (days)**	8.5 [2.5; 14]	8 [3; 16]	0.74
**Duration of ICU stay (days)**	7 [4; 17]	8 [4; 16]	0.23
**Duration of hospital stay (days)**	17 [9; 36]	15 [9.5; 27]	0.15
**Mortality at D60**	28 (17.8)	356 (29.9)	<0.01

* Organ transplant, AIDS, non-AIDS HIV, corticoids > 1 month or >2 mg/kg/day, chemotherapy, aplasia, or other immunodepression. ^§^ linezolid, daptomycin, vancomycin, cefazolin, or penicillin. IQR: interquartile; SOFA: sequential organ failure assessment; SAPS: simplified acute physiology score; ECMO: extracorporeal membrane oxygenation; MSSA: methicillin-susceptible *Staphylococcus aureus*; MRSA: methicillin-resistant *Staphylococcus aureus*; VAP: ventilation-associated pneumonia.

**Table 3 biomedicines-10-02646-t003:** Microbiological description of causal agents identified in respiratory bacterial co-infections on admission.

	All				Community-Acquired				Hospital-Acquired			
	All	Flu	Cov	*p*	All	Flu	Cov	*p*	All	Flu	Cov	*p*
**Number of infections**	135	36	99		88	31	57	.	47	5	42	.
**Gram-positive cocci**	64 (47.4)	16 (44.4)	48 (48.5)	0.68	43 (48.8)	13 (41.9)	30 (52.6)	0.34	21 (44.6)	3 (60)	18 (42.9)	0.47
** *Streptococcus pneumoniae* **	20 (14.8)	8 (22.2)	12 (12.1)	0.14	12 (13.6)	6 (19.4)	6 (10.5)	0.25	8 (17)	2 (40)	6 (14.3)	0.15
** *Staphyloccus aureus* **	35 (26)	7 (19.4)	28 (28.3)	0.30	28 (31.8)	6 (19.4)	22 (38.6)	0.06	7 (14.8)	1 (20)	6 (14.3)	0.73
** *Enterococcus* ** **sp.**	3 (2.2)	0 (0)	3 (3)	0.29	1 (1.2)	0 (0)	1 (1.8)	0.46	2 (4.2)	0 (0)	2 (4.8)	0.62
** *Moraxella catarrhalis* **	4 (3)	1 (2.8)	3 (3)	0.94	2 (2.2)	0 (0)	2 (3.5)	0.29	2 (4.2)	1 (20)	1 (2.4)	0.07
**Gram-negative bacilli**	67 (49.6)	11 (30.6)	56 (56.6)	<0.01	41 (46.6)	11 (35.5)	30 (52.6)	0.12	26 (55.4)	0 (0)	26 (61.9)	<0.01
** *Haemophilus* **	25 (18.6)	10 (27.8)	15 (15.2)	0.09	21 (23.8)	10 (32.3)	11 (19.3)	0.17	4 (8.6)	0 (0)	4 (9.5)	0.47
**Enterobacteriaceae**	43 (31.8)	6 (16.7)	37 (37.4)	0.02	21 (23.8)	5 (16.1)	16 (28.1)	0.21	22 (46.8)	1 (20)	21 (50)	0.20
**Group 1 or 2 enteric bacteria**	26 (19.2)	2 (5.6)	24 (24.2)	0.01	13 (14.8)	1 (3.2)	12 (21.1)	0.02	13 (27.6)	1 (20)	12 (28.6)	0.69
** *Proteus* **	2 (1.4)	1 (2.8)	1 (1)	0.45	2 (2.2)	1 (3.2)	1 (1.8)	0.66	0			.
** *Escherichia coli* **	12 (8.8)	2 (5.6)	10 (10.1)	0.41	6 (6.8)	1 (3.2)	5 (8.8)	0.32	6 (12.8)	1 (20)	5 (11.9)	0.61
** *Klebsiella* **	11 (8.2)	0 (0)	11 (11.1)	0.04	6 (6.8)	0 (0)	6 (10.5)	0.06	5 (10.6)	0 (0)	5 (11.9)	0.41
** *Citrobacter koseri* **	2 (1.4)	0 (0)	2 (2)	0.39	0			.	2 (4.2)	0 (0)	2 (4.8)	0.62
**Group 3 enteric bacteria**	20 (14.8)	0 (0)	20 (20.2)	<0.01	7 (8)	0 (0)	7 (12.3)	0.04	13 (27.6)	0 (0)	13 (31)	0.14
** *Enterobacter* **	13 (9.6)	0 (0)	13 (13.1)	0.02	4 (4.6)	0 (0)	4 (7)	0.13	9 (19.2)	0 (0)	9 (21.4)	0.25
** *Serratia* **	6 (4.4)	0 (0)	6 (6.1)	0.13	3 (3.4)	0 (0)	3 (5.3)	0.19	3 (6.4)	0 (0)	3 (7.1)	0.54
** *Citrobacter freundii* **	1 (0.8)	0 (0)	1 (1)	0.55	0			.	1 (2.2)	0 (0)	1 (2.4)	0.73
** *Morganella* **	1 (0.8)	0 (0)	1 (1)	0.55	0			.	1 (2.2)	0 (0)	1 (2.4)	0.73
**Nonfermentative bacteria**	11 (8.2)	1 (2.8)	10 (10.1)	0.17	6 (6.8)	1 (3.2)	5 (8.8)	0.32	5 (10.6)	0 (0)	5 (11.9)	0.41
** *Pseudomonas aeruginosa* **	7 (5.2)	1 (2.8)	6 (6.1)	0.45	4 (4.6)	1 (3.2)	3 (5.3)	0.66	3 (6.4)	0 (0)	3 (7.1)	0.54
** *Stenotrophomonas maltophilia* **	3 (2.2)	1 (2.8)	2 (2)	0.79	2 (2.2)	1 (3.2)	1 (1.8)	0.66	1 (2.2)	0 (0)	1 (2.4)	0.73
** *Acinetobacter baumannii* **	3 (2.2)	0 (0)	3 (3)	0.29	2 (2.2)	0 (0)	2 (3.5)	0.29	1 (2.2)	0 (0)	1 (2.4)	0.73
**Intracellular bacteria**	1 (0.8)	0 (0)	1 (1)	0.55	1 (1.2)	0 (0)	1 (1.8)	0.46	0			.
**Drug-resistant bacteria**	49 (36.2)	20 (55.6)	29 (29.3)	<0.01	34 (38.6)	17 (54.8)	17 (29.8)	0.02	15 (32)	3 (60)	12 (28.6)	0.15
**Extended-spectrum beta-lactamase**	5 (3.8)	0 (0)	5 (5.1)	0.17	1 (1.2)	0 (0)	1 (1.8)	0.46	4 (8.6)	0 (0)	4 (9.5)	0.47
**Carbapenemase**	0				0				0			
**AmpC-hyperproduction**	0				0				0			
**Resistant *Pseudomonas aeruginosa***	2 (1.4)	1 (2.8)	1 (1)	0.45	2 (2.2)	1 (3.2)	1 (1.8)	0.66	0			
**Methicillin-resistant *Staphyloccus aureus***	2 (1.4)	1 (2.8)	1 (1)	0.45	2 (2.2)	1 (3.2)	1 (1.8)	0.66	0			.
**More than one pathogen**	15 (11.2)	1 (2.8)	14 (14.1)	0.06	9 (10.2)	1 (3.2)	8 (14)	0.11	6 (12.8)	0 (0)	6 (14.3)	0.37

Flu: *influenza*; Cov: SARS-CoV-2.

## Data Availability

Data can be provided upon request to the corresponding author.

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
