# Peer review of "Bacterial Pulmonary Co-Infections on ICU Admission: Comparison in Patients with SARS-CoV-2 and Influenza Acute Respiratory Failure: A Multicentre Cohort Study"

_biomedicines, 2022, doi:10.3390/biomedicines10102646_

Round 1
Reviewer 1 Report
In this MS, the Authors analyzed the bacterial pulmonary co-infections at ICU admission and compared them in patients with Sars-Cov2 and influenza acute respiratory failure. The Authors claimed that data on the comparison of respiratory bacterial co-infections between patients with Influenza pneumonia and those with Sars-Cov2 pneumonia are currently scant. Therefore, the data obtained are sure to be useful to the scientific and medical communities.
The Reviewer feels it can be accepted after some minor amendments.
Comments:
In my opinion, the introduction section should be shortened (example here: 10.3390/biomedicines10081952). Detailed statistics are unnecessary here. The authors need to encourage readers to read the article, so the detailed comparison should be moved to the discussion section.
Line 133: methicillin-resistant without italics
Lines 133-134: extended-spectrum beta-lactamase (ESBL)-producing
I suggest that the comparisons in the discussion section be expanded more and include other publications - examples: 10.1186/s13613-020-00736-x, 10.3390/biomedicines10081952, 10.3390/antibiotics11070894, 10.1371/journal.pone.0251170, 10.1007/s15010-021-01661-2
Minor errors: e.g., scarce instead of scarse, line 349 (Sars-Cov2). Once Sars-Cov2 and Influenza are in italics and once not. Please correct.
Author Response
In my opinion, the introduction section should be shortened (example here: 10.3390/biomedicines10081952). Detailed statistics are unnecessary here. The authors need to encourage readers to read the article, so the detailed comparison should be moved to the discussion section.
OK, thanks for the excellent remark, we have now shortened the introduction and removed the detailed statistics.
Line 133: methicillin-resistant without italics: done
Lines 133-134: extended-spectrum beta-lactamase (ESBL)-producing: ok, done
I suggest that the comparisons in the discussion section be expanded more and include other publications - examples: 10.1186/s13613-020-00736-x, 10.3390/biomedicines10081952, 10.3390/antibiotics11070894, 10.1371/journal.pone.0251170, 10.1007/s15010-021-01661-2
Thanks for this advice. We now give more details into our discussion and have added the proposed references. All the proposed articles are now reported in table S1 and their main results are discussed into the manuscript.
Minor errors: e.g., scarce instead of scarse, line 349 (Sars-Cov2). : ok, done
Once Sars-Cov2 and Influenza are in italics and once not. Please correct.
All “Influenza” and “SARS-CoV2” are now in italics
Reviewer 2 Report
The manuscript entitled “ Bacterial pulmonary co-infections at ICU admission: comparison in patients with Sars-Cov2 and influenza acute respiratory failure: a multicentre cohort study” provides information about the impact of bacterial pulmonary co-infection (RespCoBact) during COVID-19 (CovRespCoBact). The study and the experiments are well designed. However, in order to improve the manuscript so it can be more impactful for the field, I have minor suggestions for the authors
Line 2 Sars-Cov2 should be SARS-CoV2. Please correct this throughout the manuscript.
Line 63 The complete name of the organism should be provided when it appears the first
time in the manuscript.
Line 349 ‘Sar-Cov2’ should be ‘SARS-CoV2’
Author Response
Line 2 Sars-Cov2 should be SARS-CoV2. Please correct this throughout the manuscript.: ok, we made the proposed corrections into the manuscrit
Line 63 The complete name of the organism should be provided when it appears the first : ok, done
time in the manuscript.: ok done
Line 349 ‘Sar-Cov2’ should be ‘SARS-CoV2’: ok done